# Two-to-One Trigger Mechanism for Event-Based Environmental Sensing

**DOI:** 10.3390/s25134107

**Published:** 2025-06-30

**Authors:** Nursultan Daupayev, Christian Engel, Sören Hirsch

**Affiliations:** Department of Engineering, Brandenburg University of Applied Sciences, Magdeburger Str. 50, 14770 Brandenburg an der Havel, Germany; christian.engel@th-brandenburg.de (C.E.); soeren.hirsch@th-brandenburg.de (S.H.)

**Keywords:** structural health monitoring, multivariate sensors, event-driven activation, data reduction, energy efficiency

## Abstract

Environmental monitoring systems often operate continuously, measuring various parameters, including carbon dioxide levels (CO_2_), relative humidity (RH), temperature (T), and other factors that affect environmental conditions. Such systems are often referred to as smart systems because they can autonomously monitor and respond to environmental conditions and can be integrated both indoors and outdoors to detect, for example, structural anomalies. However, these systems typically have high energy consumption, data overload, and large equipment sizes, which makes them difficult to install in constrained spaces. Therefore, three challenges remain unresolved: efficient energy use, accurate data measurement, and compact installation. To address these limitations, this study proposes a two-to-one threshold sampling approach, where the CO_2_ measurement is activated when the specified *T* and RH change thresholds are exceeded. This event-driven method avoids redundant data collection, minimizes power consumption, and is suitable for resource-constrained embedded systems. The proposed approach was implemented on a low-power, small-form and self-made multivariate sensor based on the PIC16LF19156 microcontroller. In contrast, a commercial monitoring system and sensor modules based on the Arduino Uno were used for comparison. As a result, by activating only key points in the *T* and RH signals, the number of CO_2_ measurements was significantly reduced without loss of essential signal characteristics. Signal reconstruction from the reduced points demonstrated high accuracy, with a mean absolute error (MAE) of 0.0089 and root mean squared error (RMSE) of 0.0117. Despite reducing the number of CO_2_ measurements by approximately 41.9%, the essential characteristics of the signal were saved, highlighting the efficiency of the proposed approach. Despite its effectiveness in controlled conditions (in buildings, indoors), environmental factors such as the presence of people, ventilation systems, and room layout can significantly alter the dynamics of CO_2_ concentrations, which may limit the implementation of this approach. Future studies will focus on the study of adaptive threshold mechanisms and context-dependent models that can adjust to changing conditions. This approach will expand the scope of application of the proposed two-to-one sampling technique in various practical situations.

## 1. Introduction

There is a growing interest in the analysis of the environment and its impact on the condition of building structures. The interest is due to the fact that environmental factors have a direct impact on the condition of structures and the service life of materials. Parameters such as temperature (*T*), humidity (RH), and carbon dioxide (CO_2_) are recognized as key properties for assessing the impact on material degradation processes and the general condition of building structures [1,2]. Understanding the role of each of these parameters helps to assess their influence on structural behavior and interaction with the environment. In addition to these indoor parameters, external environmental factors, especially wind, can also significantly affect both indoor microclimates and structural performance. Wind influences natural ventilation, alters temperature and humidity distribution indoors, and applies mechanical loads on building envelopes. These effects can contribute to long-term degradation and structural responses, as shown in recent studies [3], where full-scale wind monitoring demonstrated its impact on engineering structures. *T* and RH indicate the thermodynamic characteristics of the environment, reflecting the distribution of heat and steam exchange, and CO_2_ is a key indicator of air quality or gas accumulation in buildings, since its constant concentration can affect the pattern of the so-called carbonation of concrete (in case the building is made of concrete structures) [4,5]. Therefore, all these parameters are considered together, since their behavior can be interrelated. Recent studies have shown a relationship between temperature and humidity variations, especially indoors, and changes in CO_2_ levels. This correlation is mainly due to the presence of people. Therefore, temperature and relative humidity can be indirect indicators of CO_2_ dynamics in stable environments, such as indoor or poorly ventilated spaces [6]. For example, increased humidity can indicate increased dampness, insufficient ventilation, or a problem with thermal insulation leading to mold formation. An increase in CO_2_ can indicate occupancy or poor ventilation in the room, and a decrease in temperature can indicate a loss of structural integrity or a sign of poor heating system performance [7,8]. To collect and interpret such environmental indicators, appropriate sensor technologies are required. Sensor technologies are one of the key tools for implementing SHM, offering a variety of solutions for monitoring the condition of bridges, tunnels, and other critical infrastructure facilities [9,10]. A key element of these technologies is the continuous monitoring of environmental conditions such as deformation, temperature, vibration, and gas concentration, allowing for early analysis of potential signs of material degradation, indoor gas accumulation, or ventilation failure [11,12]. Sensor systems that measure these three parameters simultaneously typically require a continuous power source. In particular, CO_2_ measurements require longer measurement times, since accuracy depends on a short calibration period [13]. In contrast, temperature and humidity sensors typically offer faster response times and are often considered to be basic parameters in monitoring systems [14]. Therefore, stable and continuous measurement of all these parameters becomes particularly important in the context of structural health monitoring, where long-term environmental trends need to be accurately monitored to identify potential risks to structures [15]. Another important issue is the efficient processing of sensor data, beyond measurement accuracy. To ensure the reliability of such sensor systems, not only the detection mechanism but also data management is of critical importance. Evenly distributed use of memory for storing sensor data is critical, as memory overflow can lead to poor performance and, in some cases, hardware failure [16]. Additionally, the simultaneous processing of large amounts of data increases the computational load, resulting in higher power consumption, which is especially critical in low-power or embedded sensor systems [17]. Moreover, data reduction in real-time measurements is another crucial issue in sensors with limited hardware resources, since acquiring data, analyzing the data quickly, and storing important points at the same time requires significant computational effort. In this regard, several studies have applied data compression techniques that involve data collection, filtering, signal sampling, and removal of unimportant signals [18]. All of these challenges are largely due to the inefficiency of traditional sensor systems. Traditional sensor systems operate on a fixed time base, where all parameters are measured at fixed intervals, regardless of whether a specific event has occurred in the environment or not. Although such a process ensures the integrity of the measured data, it results in unnecessary measurements, increased storage space, and depletion of the power supply (e.g., batteries). Many studies rely on cloud-based systems, where the hardware of the sensor system is processed and the data are stored in the cloud [19,20], but this is difficult to implement in the context of low-power sensors. To optimize data storage, a data reduction method has been proposed [21]. Machine learning methods have also been proposed, but they require high computing power to train and execute on big data. In this context, recent studies have also shown the ability to reduce power consumption in sensor network systems by activating a master sensor, the so-called agent, which gives the activation command to other sensors [22]. However, even the passive state of sensors leads to constant energy consumption, especially if the sensor includes several parameters such as humidity, CO_2_, gas, and temperature. To address this issue, we propose a method aimed at reducing unnecessary dimensions by implementing an adaptive sampling strategy. Specifically, we propose a method that uses temperature and humidity to selectively activate the CO_2_ parameter. The objective is to avoid constant response to the environment and trigger a measurement only when either parameter changes above a predefined threshold. First, this approach reduces the frequency of temperature and humidity measurements based on the threshold, resulting in energy savings. Second, it is particularly suitable for resource-constrained sensors. The method was implemented as an algorithm and integrated into our custom multivariate sensor. Unlike many existing methods, this approach eliminates the need for machine learning, which is often impractical in conditions of limited hardware resources. The results show that a significant reduction in humidity measurements can be achieved without losing critical information. This contributes not only to data efficiency but also to extending the lifetime of battery-powered sensor devices. Although the proposed method shows positive expectations in certain indoor conditions, it may be limited by external factors such as human presence, window opening, and ventilation behavior. These contextual variables may change the dynamics between *T*, RH, and CO_2_, which requires further research to confirm that the approach can be effectively adapted to different environmental conditions. Future research will explore adaptive threshold mechanisms capable of dynamically adjusting activation criteria in real time. This may enable scalable deployment of this method in a variety of structural and environmental settings. Three different sensor were used in this study, which are referred to below as follows: Sensor 1—a commercial Loxone monitoring system; Sensor 2—a custom-designed module based on a PIC16LF19156 microcontroller; Sensor 3—an Arduino Uno-based platform with DHT22, DS18B20, and MQ-135 sensors.

## 2. Materials and Methods

This section describes the experimental materials, sensor types, the proposed approach, and the evaluation of the methods. First, existing data optimization methods are assessed. Then, the three sensor type (Sensor 1, Sensor 2, and Sensor 3) are described in detail, focusing on their configurations and data acquisition procedures. Then, the selected environmental conditions and sensor installation locations are described. Next, the logic of the proposed approach as an event-based measurement method is presented, as well as the rationale for activating CO_2_ sampling based on humidity (RH) and temperature (*T*) thresholds. Finally, the data processing and signal reconstruction evaluation strategy is explained, including the mean absolute error (MAE) metric. The following naming convention is adopted throughout the study: The following naming convention is adopted throughout the study:Sensor 1: Custom-developed low-power sensor based on the PIC16LF19156 microcontroller.Sensor 2: Arduino Uno-based prototype sensor platform.Sensor 3: Commercial Loxone smart monitoring system.

### 2.1. Baseline Methods for Data Reduction

Before starting the experiments, existing time series simplification methods were reviewed and tested to create a performance baseline, which allowed us to subsequently evaluate what reduction could be achieved using our method. The performance of each method was assessed by comparing the reduced and original signals (R) using the mean absolute error (MAE) as a measure of reconstruction accuracy, with the results shown in Table 1. For example, when applying the Ramer–Douglas–Peucker (RDP) [23] algorithm to our collected data of approximately 3000 data points, only 960 points were retained after simplification, resulting in a reduction of approximately 68%.(1)$$R=\left(1-\frac{N_{\mathrm{reduced}}}{N_{\mathrm{original}}}\right)\times 100\%=68\%,$$
where $N_{\mathrm{original}}$ is the total number of data points before reduction and $N_{\mathrm{reduced}}$ is the number of data points after the reduction method. To assess the accuracy of the reconstruction, we interpolated the reduced signal back to the original time axis and calculated the mean absolute error (MAE) between the original and reconstructed signals.(2)$${\mathrm{MAE}}_{\mathrm{RDP}}=\frac{1}{n}\sum_{i=1}^{n}\left|x_{i}-{\hat{x}}_{i}\right|=0.0091$$
where $x_{i}$ is the original value, ${\hat{x}}_{i}$ is the reconstructed value at index *i*, and *n* is the total number of observations. Using this metric, each method in the table was applied to the same set of sensor data. According to our observations, those with higher MAE and lower data reduction rates were not included in the table. As a result, this review allowed us to determine how each approach can recover from reduction or whether it is possible to use it as a hybrid with our approach, especially in the context of low-power sensors. For example, principal component analysis (PCA) can be used as a complement to reduce the dimensionality of data in memory, thereby increasing storage capacity [24]. In general, combining several methods with our proposed approach may be the next step in implementation, especially when developing on more powerful microcontrollers (for example PIC16LF19186 [25]).

These data compression methods are discussed to provide context for the proposed approach. However, many of the listed methods are widely used for signal reduction and compression techniques in sensor networks, but most of them require significant computational resources, memory, or complex pre-processing pipelines, making them less suitable for real-time execution on ultra-low-power microcontroller-based platforms. For improving sensor performance and data quality, they often require significant computational and memory resources. In contrast, our method is designed specifically for microcontroller-based systems with tight energy and memory constraints. The 2-to-1 sampling strategy avoids complex computations by relying on threshold-based event triggering, making it more suitable for lightweight embedded applications. In contrast, the proposed method in this study is designed specifically for constrained environments. It enables sampled data collection with minimal computation and without the need for prior transformation or training, making it more suitable for embedded sensor systems with low memory power.

### 2.2. Sensor Setup and Experimental Environment

The initial objective of this study was to find a suitable location to ensure proper analysis from different locations and to place sensors to collect environmental data. The installation locations were chosen based on typical room air circulation patterns, wall accessibility, and room usage to capture realistic and dynamic changes in CO_2_, temperature, and humidity. This setup allowed each sensor to represent different conditions such as high occupancy, low occupancy, and stable environments. For this selection, three types of places were selected, those located away from the city (with minimal influence of urban factors such as road vibration and noise) (see Figure 1) and rooms within buildings located inside the city, where one room has a frequent presence of people while the other is often empty (see Figure 2).

These conditions allowed us to compare different climate parameters under varying levels of external influence. All measurements were carried out over a period of one month, with sensors placed between two structural elements in the house and others in the corner of a wall inside the building. This arrangement allowed us to record changes in *T*, *H*, and CO_2_, including increases, decreases, and sharp fluctuations. As part of the monitoring setup, we used the commercial smart system Loxone, which was configured to measure three parameters simultaneously: temperature, humidity, and CO_2_ concentration.

The next sensor, designed specifically for integration in hard-to-reach places, also measures three parameters simultaneously (see Figure 3). This sensor is based on the PIC16LF19156, has built-in flash memory for local data logging, and has an optional connectivity module. The connectivity components include a digital temperature and humidity sensor and a non-dispersive infrared CO_2_ sensor. The sensor was originally designed for use in industrial environments where energy-efficient operation and modular integration are important. The third measuring device was a simple one based on Arduino Uno, where separate RH, *T*, and CO_2_ modules were connected. All modules were connected to a breadboard, where the code was subsequently written to measure the environment and the data were stored in a built-in flash card module.

Each sensor consists of several key components, each of which has specific parameters for measuring the environment. The PIC16LF19156 microcontroller offers low power consumption, compact size, and essential peripheral modules, such as ADC, UART, and EEPROM. A component based on the HS4111 was used to measure *T* and *H*, operating in the range of −10 °C to +80 °C with an accuracy of ±0.2 °C for temperature, and from 10% to 90% RH with an accuracy of ±1.5%. A 12-bit ADC converter was used for low-noise signal acquisition, buffered by a 10 MHz analog front end. CO_2_ levels were measured using an STC-C4 sensor (400 to 5000 ppm, accuracy ±100 ppm). The module is equipped with a UART communication interface and integrates with the PIC16LF19156 via serial connection. A real-time clock (RTC) was added to support consistent CO_2_ readings. Additional components included a 3-axis accelerometer (ADXL325, ±5 g) and a photodiode (SFH5711) for light-level detection. The architecture is modular, with 6 analog channels and I2C communication for expansion. An Arduino-based system included the DHT11 and DS18B20 sensors for humidity and temperature, respectively, and an MQ-135 gas sensor for CO_2_ detection. The DS18B20 measures from −15 °C to +50 °C with typical accuracy of ±0.5 °C. The MQ-135 detects gases like NH_3_, alcohol, and CO_2_. All modules were connected to an Arduino Uno, and real-time data were collected via serial monitoring. The Loxone sensor system operates from –20 °C to +55 °C and handles up to 95% relative humidity. It connects via the Loxone Tree interface for integration with the Miniserver or Tree Extension. The system measures temperatures from –40 °C to +120 °C (±0.5 °C) and humidity from 0% to 100% RH (±2%). For CO_2_ monitoring, it covers a range of 400–10,000 ppm with an accuracy of ±(30 ppm + 3%). All systems recorded environmental parameters every 15 min. In the PIC-based sensor, *T* and RH were always measured, while CO_2_ was measured selectively using the following trigger logic:$$\mathrm{trigger}\left(t\right)=\left\{\begin{array}{cc} 1 & \mathrm{if}\;|T\left(t\right)-T\left(t-1\right)|>0.5\;{}^{\circ}C\;\mathrm{and}\;\left|RH\left(t\right)-RH\left(t-1\right)\right|>2\%\\ 1 & \mathrm{if}\;t\;\mathrm{mod}\;20=0\\ 0 & \mathrm{else} \end{array}\right.$$

Data were stored in CSV format with headers: “timestamp, temperature, humidity, CO_2_”. The embedded C code was written in MPLAB X IDE v6.20 and compiled using the XC8 toolchain. The UART was configured for 9600 baud transmission. Power was supplied by a 3.7 V, 1200 mAh Li-Po battery with an MCP1700 regulator. The average current consumption was 1.8 mA (sleep) and 19 mA (active sensing). Signal reconstruction accuracy was evaluated using the MAE and RMSE metrics in Python (version 3.10), with the full-resolution Loxone data used as the reference.

### 2.3. Data Acquisition

In the data acquisition stage, data collection was performed in parallel using three sensors, each programmed to measure all three parameters—temperature, humidity, and CO_2_—at intervals ranging from 1 to 15 min. The observation period lasted for one month (Sensor 1 and Sensor 3: March; Sensor 2: April), enabling the recording of a complete time profile of environmental parameter changes. For example, in Sensor 3, data were saved to memory (SD card), while in Sensor 2, data were transmitted directly via USB. Although Sensor 2 recorded data in April, this time window was intentionally chosen to closely match the external conditions observed in March. Factors such as ambient temperature, room usage, and ventilation schedules were held constant to ensure comparability. The test environment remained stable between the two months, allowing for a fair comparison of performance across all platforms. In Sensor 1, data were written to the internal memory of the controller. All collected values were subsequently visualized, allowing for an analysis of the relationships between the parameters. To provide a clearer understanding of how the measurements were obtained and structured, the data collection process is described in more detail below. As for Sensor 1, the data were automatically exported in CSV format, and owing to its accuracy and stability, Sensor 1 data did not require additional pre-processing before time-series analysis (see Figure 4).

As for Sensor 3, the data collection program was written using the Arduino IDE with libraries such as DHT.h, OneWire.h, and MQ135.h to communicate with the sensors. Data readings were collected every 15 min, and saved to the SD card, where they were later stored in CSV format. It is worth noting here that the data were also slightly processed using Python (version 3.10), which included libraries such as pandas, numpy (version 2.3.1), and matplotlib (version 3.10.1) for pre-processing and visualization (see Figure 4).

Simultaneously, data were collected using a custom-designed sensor based on a PIC16LF19156 microcontroller and a program written in MPLAB X IDE using the Microchip Code Configurator (MCC), with a particular focus on three types of sensors: a humidity sensor, a digital temperature sensor (e.g., DS18B20), and a STCC4-based CO_2_ sensor. The resulting data were time-stamped and missing values were handled using direct interpolation. Matplotlib (version 3.10.1), seaborn (version 0.13.2), and scipy (version 1.14.0) were used for data analysis (see Figure 5).

The proposed method was implemented solely on a PIC16LF19156-based sensor, but comparisons with Arduino and Loxone systems were included as a basis for performance and realism. The Arduino-based setup is a typical low-cost prototype without data reduction, while the Loxone system is a commercial smart environmental monitoring solution with continuous measurement. By collecting data from all three platforms under the same environmental conditions, we were able to confirm the reliability of the PIC-based measurements and provide a benchmark for the signal shape and dynamics. This comparative design allowed us to evaluate the trade-off between sampling reduction and signal accuracy by reconstructing the reduced signal from the PIC device and comparing it with the full-resolution data from the Loxone system.

### 2.4. Correlation

A critical component of the methodology involves analyzing the relationships between the environmental parameters. The goal is to determine whether *T* and RH contain enough information about the behavior of CO_2_ to act as triggers or whether *T* and CO_2_ have a better relationship to be used as triggers for one parameter. For this purpose, a correlation analysis was applied to the parameters using the data collected from three sensors. It is necessary to determine how consistently changes in temperature and humidity are associated with fluctuations in the concentration of CO_2_ and how reliable these two parameters can be without regularly measuring CO_2_. If a pronounced correlation is detected, then the two parameters can serve as a reliable basis for predicting the behavior of the third parameter.

Based on the correlation matrix (see Figure 6), the data obtained from Sensor 2 demonstrate a weak positive correlation between *T* and CO_2_ (r=0.34) and between RH and CO_2_ (r=0.37). According to the data from Sensor 3, there is a strong correlation between *T* and CO_2_ (r=0.39), while the correlation between RH and CO_2_ is also positive but weaker (r=0.24). However, *T* and RH are practically unrelated to each other (r=0.03). The dataset from Sensor 1, however, represents a positive correlation between *T* and RH (r=0.53), but CO_2_ is practically independent of both parameters (r=−0.18 with *T* and r=0.02 with RH). As a result, two of the sensors revealed a positive correlation between temperature, humidity, and CO_2_ concentration. This allows these parameters to be used as a basis for developing trigger logic for activating the measurements.

For further confirmation of the correlation analysis, a combined pairplot was constructed (see Figure 7), illustrating the distribution and interrelation of the parameters from each sensor’s dataset. This is particularly important when constructing the trigger activation logic. The pairplot shows that temperature and humidity in Sensor 1 and Sensor 2 data are concentrated within a narrow range, while Sensor 3 demonstrates a more uniform distribution. CO_2_ in all three cases exhibits a pronounced right-hand shift, indicating the passive presence of people in the premises. The data from Sensor 2 and Sensor 3 show a positive trend between temperature and CO_2_, which is consistent with the previously obtained correlation matrix. The data from Sensor 1, however, show less pronounced relationships between temperature and CO_2_, with a chaotic scattering of points, confirming a weak correlation. Although the weak correlation observed in Sensor 1 may indicate a lack of coupling between T, RH, and CO_2_, it is important to clarify that all three sensor systems were deployed under identical environmental conditions, using synchronized sampling intervals (15 min) and comparable locations. The weaker coupling in Sensor 1 is likely due to limited sensitivity or accuracy of the sensor hardware, rather than differences in environmental exposure or measurement timing. Moreover, the proposed 2-to-1 sampling logic does not rely on inter-device correlation, but instead operates based on intra-sensor threshold exceedance. Therefore, even if global correlations appear weak in a single sensor, local variations within a single data set are sufficient to trigger CO_2_ measurements. This design improves the robustness and adaptability of the method to different sensor configurations and environments.

### 2.5. Trigger Justification

As a further step in the analysis of the influence of temperature and humidity on the CO_2_ levels, a two-dimensional heat map (Figure 8) was constructed based on the partitioning of the two parameters into discrete intervals. The horizontal axis represents the temperature intervals and the vertical axis the humidity intervals. The color of each cell shows the average value of CO_2_ concentration for a combination of temperature and humidity values. According to the data from Sensor 3, CO_2_ exceeds 1000 ppm in the region corresponding to temperatures of 21–22.5 °C and humidity levels of 37–41%. This indicates that increasing temperature and humidity are suitable triggers for CO_2_ measurement. Similarly, the data from Sensor 2 show a stable trend of CO_2_ increasing with increasing temperature (>21 °C) and humidity. In contrast, the data from Sensor 1 show a very weak correlation, though temperature and humidity may still be used as a combined trigger for CO_2_ sampling. This discrepancy can be explained by differences in sensor configuration, placement, or environmental conditions. In particular, Sensor 1 may have experienced different airflow conditions due to its position in the room. These factors likely contributed to the reduced correlation strength. Therefore, the observed weak relationship should not be interpreted as a failure of the proposed 2-to-1 logic, but rather as a reflection of platform-specific influences.

Building on this observation, the main proposed approach, which is based on the “two-to-one” principle, involves activating one parameter (in this case CO_2_) based on the behavior of the other two (*T*, RH). Within this pattern, two parameters are monitored simultaneously and if the change in each of them exceeds a set threshold or a checkpoint in time is reached, then the third parameter is measured. To formalize this concept, we define three parameters as $x_{1}\left(t\right)$: *T* at time *t*, $x_{2}\left(t\right)$: RH at time *t* and y(t): CO_2_. $\theta_{1}$, $\theta_{2}$ are the threshold values for $x_{1}$ and $x_{2}$. $\Delta x_{i}\left(t\right)=\left|x_{i}\left(t\right)-x_{i}\left(t-1\right)\right|$ represent the absolute change in parameter $x_{i}$ at time *t*. k∈N is a fixed control interval (e.g., every 20 time steps). The conditional activation rule for measuring *y* is defined as(3)$${\mathrm{Trigger}}_{y}\left(t\right)=\left\{\begin{array}{cc} 1, & \mathrm{if}\;\left(\Delta x_{1}\left(t\right)>\theta_{1}\wedge \Delta x_{2}\left(t\right)>\theta_{2}\right)\vee \left(t\;\mathrm{mod}\;k=0\right)\\ 0, & \mathrm{else} \end{array}\right.$$
where ${\mathrm{Trigger}}_{y}\left(t\right)=1$ means that at time step *t*, a new measurement of y(t) is executed. The logic behind the 2-to-1 threshold activation is summarized in the flowchart below Figure 9, illustrating the decision-making steps in the sensor firmware.

Although the proposed 2-to-1 triggering logic was only implemented on a PIC16LF19156-based sensor, the inclusion of the Arduino and Loxone platforms in this study served to establish reference baselines. These two systems, operating in continuous measurement modes, allowed for an indirect validation of the reconstructed CO_2_ signal obtained using the proposed triggering logic. Although the same threshold logic could not be directly implemented on the Loxone due to firmware limitations and platform constraints, the comparative analysis allowed for a meaningful assessment of signal preservation, measurement accuracy, and power tradeoffs. Also, the implementation of similar threshold logic on Arduino is possible; in practice, it requires significant firmware modifications and does not provide the required level of energy efficiency and stability without additional optimization. Therefore, instead of directly implementing the algorithm on these platforms, a comparative measurement was performed to analyze the accuracy and shape of the signal. This approach allowed us to objectively evaluate the advantages of the proposed activation logic under conditions of limited computing and energy resources.

### 2.6. Threshold Selection

After two parameters are approved as trigger parameters for the activation of the third parameter measurements, the next process is to determine the threshold values. A grid search was used to select the thresholds, and the *Precision*, *Recall*, and the *F1 Score* quality metrics were computed for each combination. Precision measures how many of the triggered events were true for CO_2_ peaks, Recall measures how many actual CO_2_ peaks were correctly detected, and the F1 Score is the harmonic mean of precision and recall used to balance the two metrics:(4)$$\mathrm{Precision}=\frac{TP}{TP+FP}$$(5)$$\mathrm{Recall}=\frac{TP}{TP+FN}$$(6)$$F_{1}=\frac{2\cdot \mathrm{Precision}\cdot \mathrm{Recall}}{\mathrm{Precision}+\mathrm{Recall}}$$

Based on the grid search described above, we systematically evaluated all relevant combinations of temperature and humidity thresholds. The combination that achieved the highest F1 Score was selected as the optimal trigger condition, and the following thresholds were determined: the temperature threshold is T>20.5 °C and the humidity threshold is H>31%. This threshold pair yielded the best trade-off between Precision and Recall, achieving an F1 Score of 0.473, with a trigger rate of 95.4%. This means that most high-CO_2_ events were successfully detected, although with a relatively frequent activation rate. The selected thresholds were used throughout the trigger behavior simulation and system implementation on the PIC-based sensor platform. The top five threshold combinations including Precision, Recall, F1 Score, and triggering rates are presented in Table 2. As shown, several threshold pairs resulted in nearly identical F1 Scores.

To better visualize the overall performance, Figure 10 presents the heatmap of the F1 Score, which clearly highlights areas with the best activation accuracy.

This visualization allows us to identify threshold regions that provide optimal detection performance. Each cell in the heatmap represents an F1 Score obtained from a specific combination of temperature and humidity thresholds. Darker regions indicate combinations with higher F1 Scores, which means a better balance between Precision and Recall for detecting CO_2_ peaks.

## 3. Results

Figure 4, Figure 5, Figure 10, Figure 11, and Figure 12 together present the complete workflow of the proposed method, from raw data to downsampling and reconstruction. Figure 4 and Figure 5 present the original environmental data that serve as the basis for threshold selection. Figure 10 shows the F1 Score heatmap obtained from a grid search through the temperature and humidity thresholds, highlighting the optimal combinations for triggering CO_2_ measurements. To illustrate the activation process in more detail, Figure 11 provides an example time series of how thresholds were applied to generate CO_2_ measurement triggers. The upper part displays a sharp increase in CO_2_. The middle and lower panels display the temperature and humidity signals. The dashed lines indicate the activation thresholds (20.5 °C and 31%). The black markers indicate the moments when both thresholds are exceeded simultaneously, thereby activating the CO_2_ measurement. These thresholds were then applied to the data, and Figure 12 shows the reconstructed CO_2_ signal obtained from the reduced set of measurement points. The reconstruction was performed using linear interpolation between the activated CO_2_ measurement points, assuming a linear change between successive activated values. In the next graph, a more visual representation of the reconstructed signal from the reduced data is presented.

After triggering the CO_2_ measurements based on certain thresholds, the reduced data were used to reconstruct all parameters. The resulting reconstruction is visualized in the Figure 12, demonstrating that the main signal characteristics were preserved despite the lower sampling rate.

Figure 11 shows the behavior of the 2-to-1 triggering mechanism over time. The top panel shows the raw CO_2_ concentration signal, where a significant increase occurs, likely due to human presence or ventilation changes. The middle and bottom panels display temperature and humidity values, respectively. The dashed horizontal lines represent the fixed thresholds of T>20.5 °C and RH>31%. Black dots indicate the moments when both thresholds were exceeded simultaneously, thus triggering a CO_2_ measurement. These events align well with the sharp changes in CO_2_, validating the effectiveness of the trigger logic. Figure 12 presents the reconstructed CO_2_ signal based on the reduced sampling set. The reconstructed curve closely follows the shape of the original signal, capturing all major peaks and trends. This confirms that the event-based method retains the essential characteristics of the full-resolution signal. The overall reduction in CO_2_ measurements by 41.9% resulted in minimal loss of accuracy, as previously quantified (MAE = 0.0089, RMSE = 0.0117). This supports the claim that the proposed method can significantly reduce sensor activity while preserving signal quality in indoor environments. The proposed 2-to-1 sampling method, which activates CO_2_ measurements only when the temperature exceeds 20.5 °C and relative humidity exceeds 31%, resulted in a data reduction of 41.9%. Despite this reduction, signal reconstruction maintained high accuracy, with a mean absolute error (MAE) of 0.0089 and a root mean square error (RMSE) of 0.0117. The observed 41.9% reduction in the CO_2_ sampling rate directly contributes to energy savings, since CO_2_ sensors typically consume significantly more power than temperature or humidity sensors, especially due to their longer stabilization times. By reducing the number of activations, the method reduces the overall system energy budget. Furthermore, the implementation on a PIC16LF19156 microcontroller demonstrates that the proposed approach supports both energy efficiency and compact installation, making it highly suitable for embedded and battery powered applications.

## 4. Discussion

In this study, a threshold-based activation method for reducing CO_2_ measurements by two-to-one was proposed. The main goal of the proposed two-to-one method is to trigger the measurement of one parameter only when the thresholds of two other parameters are exceeded simultaneously. It is important to note that the trigger pair is not necessarily temperature and humidity, but can be combined depending on environmental events. In this experiment, temperature and humidity are the trigger parameters for CO_2_. To evaluate the method, three different sensors were used that measured over a two-month period (March and April). The results showed the ability to reduce the data but have several limitations. First, the analysis was conducted over a two-month period, which is a very short time to fully prove the choice of *T* and RH. For this, it is necessary to measure over the entire year in order to take into account all seasonal weather conditions, as well as changes in ventilation, the absence of people in the premises, weather variability, etc. Second, the use of static thresholds is limiting in the context of adaptive CO_2_ activations; the optimal solution would be to use a dynamic threshold mechanism that could adapt to all seasonal and temporal environmental events. Third, it is important to note that CO_2_ readings may be more effective in the summer months. Setting a smaller interval (every second) can improve the accuracy of detecting significant changes and improve the trigger response. In experiments, this resulted in a reduction in the number of CO_2_ samples by approximately 67%, which directly translates into a reduction in power consumption, since CO_2_ sensors typically require longer warm-up time compared to temperature and humidity sensors. Moreover, the entire system was implemented on a compact microcontroller platform (PIC16LF19156) with a small-form-factor sensor design measuring only 3 × 2 cm.

## 5. Conclusions

In this study, we proposed a two-to-one threshold triggering mechanism to selectively trigger CO_2_ measurements based on temperature and relative humidity changes. The method was implemented and tested on a low-power sensing platform using the PIC16LF19156 microcontroller and compared with Arduino and Loxone systems. The results showed that the CO_2_ measurements can be reduced by about 41.9% without compromising the fundamental signal performance. The reconstructed data showed a good accuracy level with MAE = 0.0089 and RMSE = 0.0117, confirming the effectiveness of the method for indoor environmental monitoring. The logic is lightweight, does not require machine learning, and is well suited for resource-constrained embedded devices, as demonstrated by the compact and power efficient PIC16LF19156 platform. The proposed method enables energy-efficient sensing strategies and is particularly suitable for long term monitoring in buildings, especially in space-constrained areas. Compared to continuous measurement systems, event-based sampling significantly reduces power consumption and data transmission frequency. The method is generalizable and can be integrated into various microcontroller-based platforms, making it scalable for broader environmental sensing applications. Future work will focus on extending the model with adaptive or seasonal thresholds, optimizing it for outdoor or industrial settings with dynamic environmental changes, and comparing it with more advanced baseline data processing methods.

## 6. Limitations

The proposed two-to-one threshold-based method demonstrates effectiveness in reducing the frequency of CO_2_ measurements and energy consumption. However, several limitations remain. First, the relationship between temperature, humidity, and CO_2_ is not universally stable. Although correlations were observed during experiments conducted in March and April, they were conducted under relatively controlled conditions and may not fully reflect the variability of dynamic indoor environments. Second, the thresholds were determined empirically, based on specific experimental observations, rather than using a generalizable model. A key limitation of the proposed method is the assumption that fixed thresholds for *T* and RH can reliably predict CO_2_ dynamics under different conditions. In reality, these thresholds may vary due to factors such as ventilation, occupancy, air conditioning, or room shape. To address this issue, future works could include a calibration phase in which local correlation statistics are collected to dynamically determine optimal thresholds. Alternatively, a method could be developed to adapt the activation logic to different environmental conditions.

## 7. Future Work

Future work will focus on improving the generalizability and adaptability of the proposed two-to-one threshold-based activation method. First, the system will be tested under a wider variety of environmental conditions, including different seasons, ventilation states, and occupancy levels, to assess its robustness and refine the trigger conditions. Second, dynamic threshold selection will be explored using either rolling statistical models or lightweight learning-based approaches to automatically adjust the activation logic based on historical trends and environmental feedback. Another plan involves extending the method to other environmental parameters or pollutants that may also benefit from selective sampling strategies. Additionally, efforts will be made to integrate energy harvesting techniques to further improve autonomy in long-term deployments. Finally, scaling the approach to multi-room or building-wide networks will be explored, including sensor coordination and adaptive threshold propagation across nodes.

## Figures and Tables

**Figure 1 sensors-25-04107-f001:**
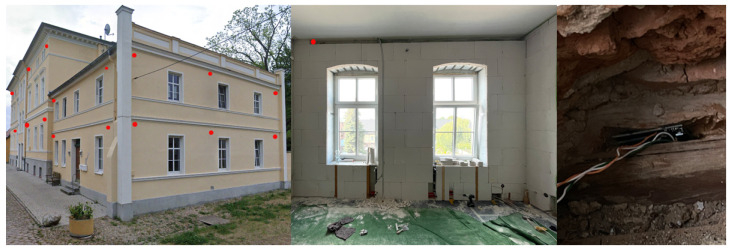
Positioning of Sensor 1 (indicated by red dot) in room corner in a historic manor house from the Wilhelminian period, adapted from Finn E. Schmid-Bonde, Fachhochschule Potsdam.

**Figure 2 sensors-25-04107-f002:**
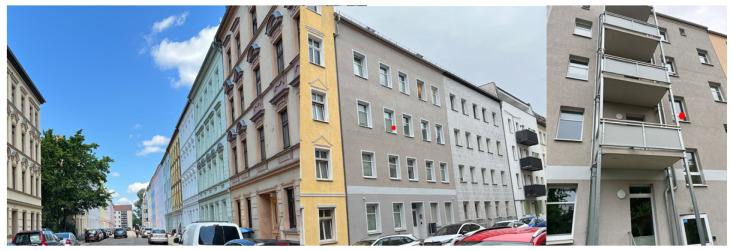
Placement of Sensors 2 and 3 in two rooms within the same building: the image shows, from left to right, the positions of the two monitored rooms.

**Figure 3 sensors-25-04107-f003:**
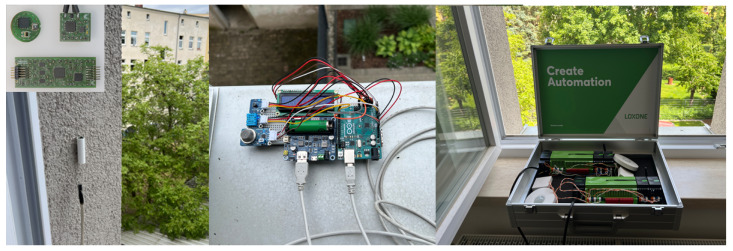
From left to right: a custom PIC16LF19156-microcontroller-based (Sensor 1), Arduino based platform (Sensor 2), and Loxone smart system (Sensor 3).

**Figure 4 sensors-25-04107-f004:**
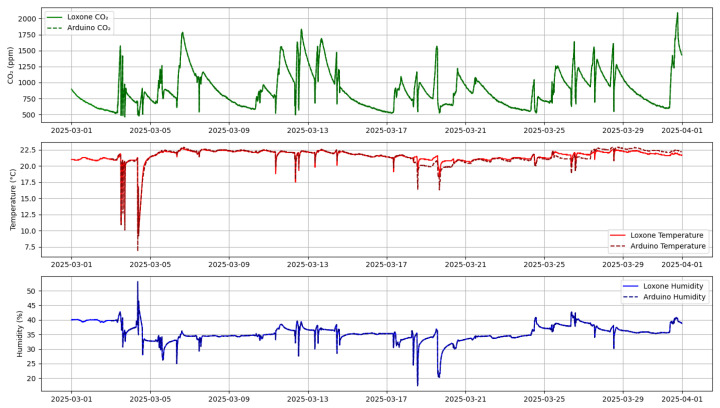
Environmental data recorded by Sensors 1 and 3 during March. Green indicates CO_2_ concentration, red indicates temperature, and blue indicates humidity.

**Figure 5 sensors-25-04107-f005:**
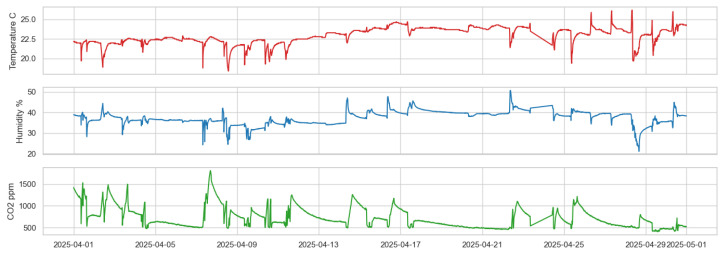
Environmental data recorded by Sensor 2 during April. Red indicates temperature, blue indicates humidity and green indicates CO_2_ concentration.

**Figure 6 sensors-25-04107-f006:**
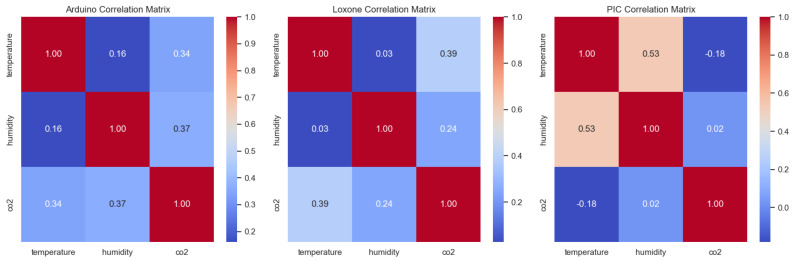
Correlation matrix of environmental parameters (*T*, RH, CO_2_) for all three sensors.

**Figure 7 sensors-25-04107-f007:**
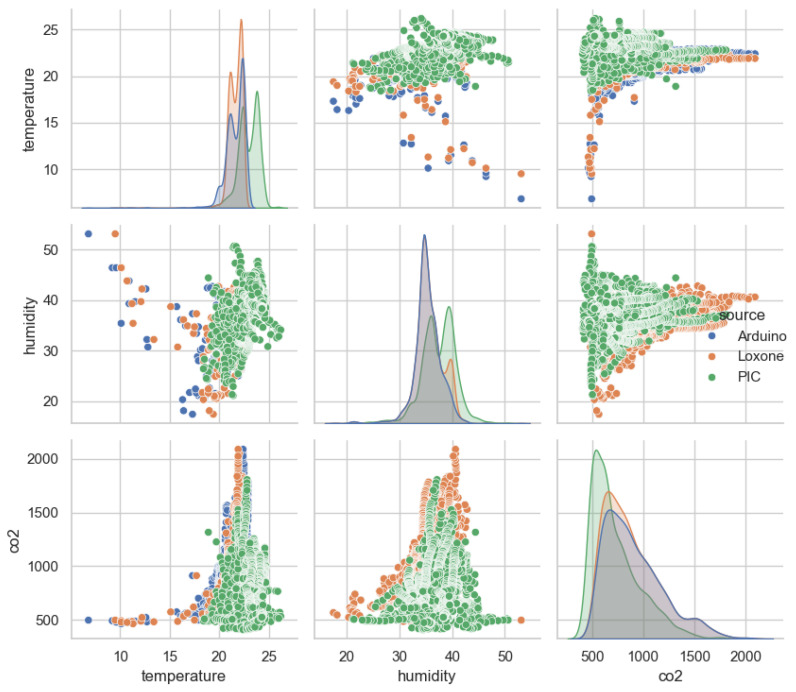
Pair plot used to evaluate the relationships between environmental parameters (temperature, humidity, CO_2_) relevant to trigger logic design. Blue represents Arduino, orange represents Loxone, and green represents PIC-based sensor system. The diagonal plots show kernel density estimates (KDE) for each parameter, while the scatter plots show pairwise relationships.

**Figure 8 sensors-25-04107-f008:**
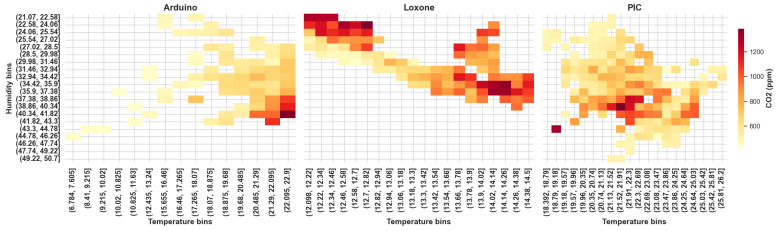
Heat map of average CO_2_ concentration across temperature and humidity intervals.

**Figure 9 sensors-25-04107-f009:**
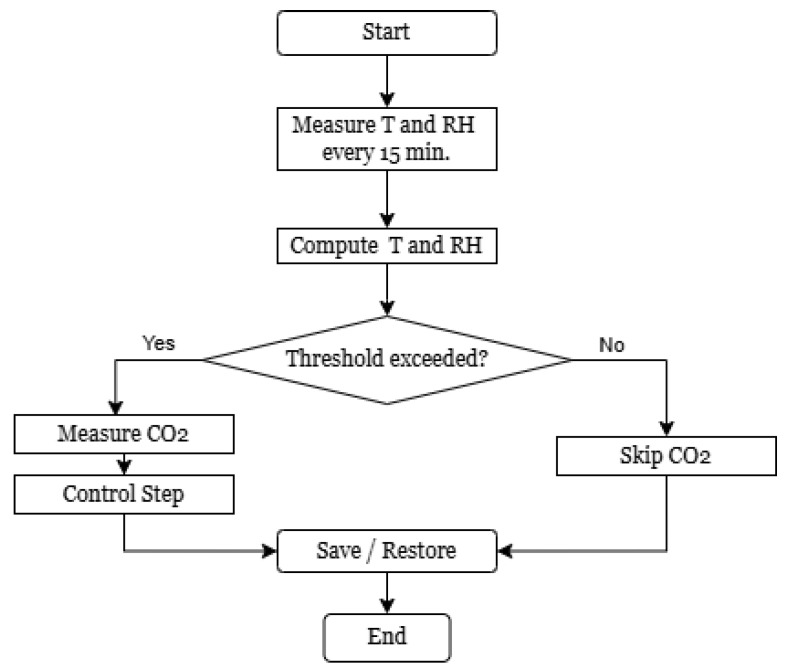
Flowchart illustrating the 2-to-1 threshold-based activation logic for CO_2_ measurement based on changes in temperature and humidity.

**Figure 10 sensors-25-04107-f010:**
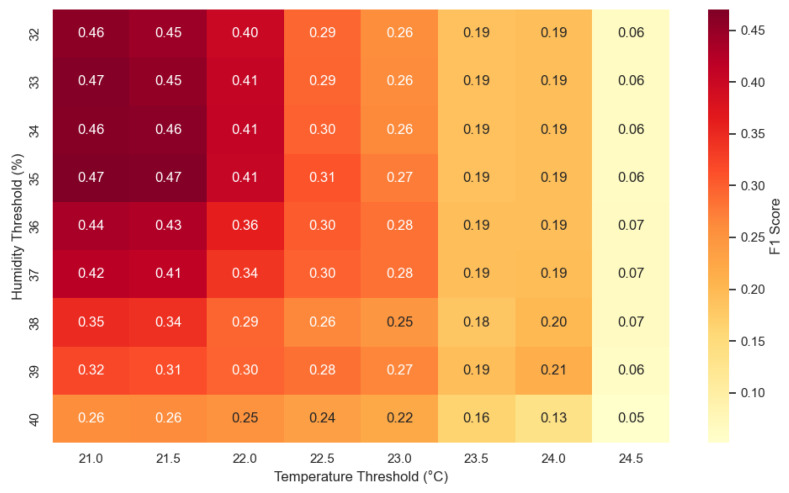
F1 Score heatmap for CO_2_ trigger thresholds.

**Figure 11 sensors-25-04107-f011:**
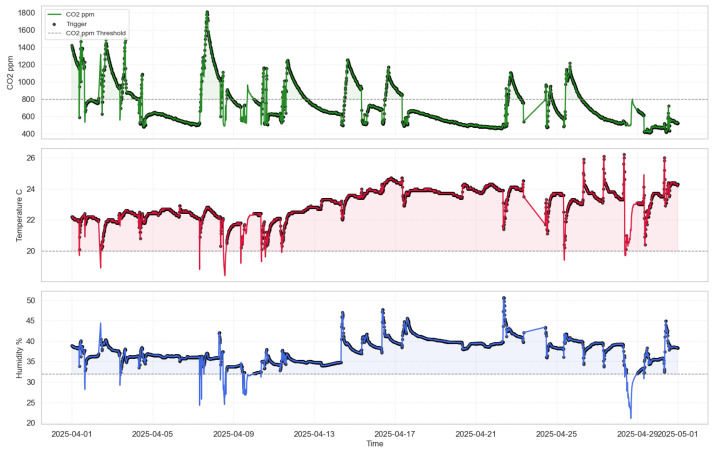
Activation of CO_2_ measurements based on threshold exceedance of temperature and humidity. The green line represents CO_2_ concentration, the red line shows temperature, and the blue line shows relative humidity. Black markers indicate activation triggers, and the dashed grey line marks the CO_2_ threshold.

**Figure 12 sensors-25-04107-f012:**
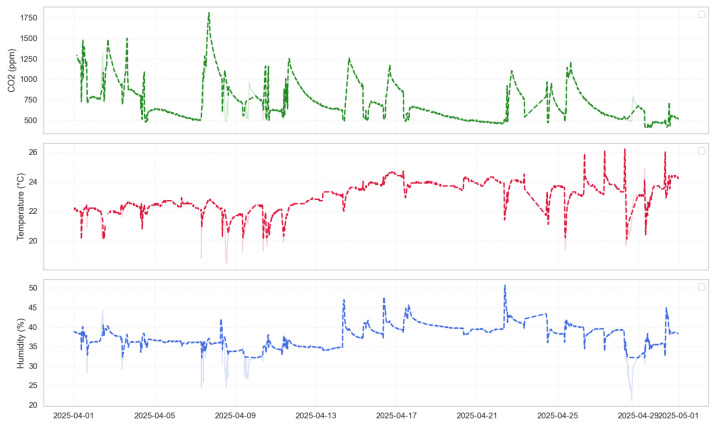
Reconstructed signals from reduced measurement points using the proposed threshold-based sampling method. The green line shows CO_2_ concentration, the red line shows temperature, and the blue line shows relative humidity.

**Table 1 sensors-25-04107-t001:** Data Reduction Methods.

Method	Description	Reduced	MAE
RDP [26]	Keeps only the key points that form the waveform.	68%	0.0091
PAA [27]	Reduces the signal dimension by dividing it into segments of equal length.	62%	0.0103
vSAX [28]	Converts PAA values to discrete characters.	65%	0.0125
Wavelet Tr. [29]	Decomposes a signal into components and removes small signal coefficients.	70%	0.0082
Kalman Filter [30]	Smoothes and reduces noise.	60%	0.0069
Compressive Sensing [31]	Reconstructs signals from fewer samples.	72%	0.0073
Delta Encoding [32]	Reduces the redundancy of similar values.	50%	0.0132
Peak-to-Peak [33]	Keeps only significant peaks and valleys of the signal.	59%	0.0108
Entropy-Based [34]	Preserves those signal segments with high information content.	64%	0.0099
Variance-Based [35]	Compresses data to a value exceeding the threshold.	61%	0.0106

**Table 2 sensors-25-04107-t002:** Top 5 threshold combinations for CO_2_ trigger (Sensor 1).

T Thresh. (°C)	H Thresh. (%)	Precision	Recall	F1	Trigger %
20.5	31	0.311	0.990	0.473	95.4
20.0	31	0.310	0.996	0.473	96.3
20.5	35	0.329	0.842	0.473	76.8
20.0	35	0.328	0.847	0.473	77.3
19.5	35	0.327	0.848	0.472	77.6

## Data Availability

Data used in this study are not publicly available but can be obtained from the authors upon request.

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
