# Peer review of "Two-to-One Trigger Mechanism for Event-Based Environmental Sensing"

_sensors, 2025, doi:10.3390/s25134107_

Round 1

Reviewer 1 Report

Comments and Suggestions for Authors This paper proposed a 2-to-1 threshold sampling approach, where the CO2 measurement is activated when the specified T and RH change thresholds are exceeded. As a result, by activating only key points in the T and RH signals, the number of CO2 measurements was significantly reduced without loss of essential signal characteristics. Signal reconstruction from the reduced points showed acceptable accuracy, as confirmed by the results of calculating MAE and RMSE. Overall, this paper has a reasonable framework, rich content, reliable results, and strong engineering significance. It can be published with minor revisions. 1. The innovation points in the abstract section are insufficient, and there is a lack of quantitative indicators to verify the quality of the methods. 2. The introduction section can add more descriptions, such as the impact of environmental winds, for example, Engineering Structures, Vol. 327, No. 3, 119523. 3. More details need to be provided in the materials and methods to facilitate the reproducibility of the results. 4. How is the installation position of the sensor determined? What is the performance of the sensor? 5. Table 3 can be represented in text. 6. A flowchart can be added to the paper to enhance its readability. 7. The content of the results section is limited and needs further refinement, such as adding explanations for Figures 11 and 12 8. The paper lacks a conclusion section.

Author Response

Comments 1: The innovation points in the abstract section are insufficient, and there is a lack of quantitative indicators to verify the quality of the methods.

Response 1:  I have included innovative point and quantitative results (sampling reduction rate, MAE, and RMSE) to show the method’s accuracy and efficiency in Abstracts Section.

Comments 2: The introduction section can add more descriptions, such as the impact of environmental winds, for example, Engineering Structures, Vol. 327, No. 3, 119523.

Response 2: I have added a description in the Introduction section (starting at “In addition to these indoor parameters, external environmental factors….”). It was also cited the recommended article.

Comments 3: More details need to be provided in the materials and methods to facilitate the reproducibility of the results.

Response 3: Section 2. Materials and Methods - starts from this text “Each sensor consists of several key components...” and ends “and I2C communication for sensor expansion.”

Comments 4: How is the installation position of the sensor determined? What is the performance of the sensor?

Response 4: The explanation was added in subsection 2.2 'Sensor Setup and Experimental Environment', beginning with : 'The installation locations were selected based on typical patterns of indoor air circulation and room occupancy.'

And the performance of each sensor, including measurement range, accuracy, operating conditions, and power consumption, are provided in Section 2.2 “Sensor Setup and Experimental Environment”.

Comments 5: Table 3 can be represented in text.

Response 5: Table 3 is described as a text. (Table removed) in Section Results beginning with “The overall reduction of $CO_2$ measurements by 41.9\% resulted in minimal loss…”

Comments 6: A flowchart can be added to the paper to enhance its readability.

Response 6: A flow chart has been added fig. 9 in section “Trigger Justification”

Comments 7: The content of the results section is limited and needs further refinement, such as adding explanations for Figures 11 and 12

Response  7:

In the Results section, explanations have been added to Figures 11 and 12 to clarify its significance. (Figure 11) The activation logic and quality of signal recovery after contraction are described (Figure 12)

Comments 8: The paper lacks a conclusion section.

Response 8: \section{Conclusion} added

Reviewer 2 Report

Comments and Suggestions for Authors

The comments are in the attached document.

Author Response

We thank the reviewer for the detailed review! You will find responses to your comments bellow:

1.a.

Comment: The work proposes a method that uses two threshold values from T and RH to trigger the measurement of CO2 to save the usage of energy, data acquisition and storage, but still be able to achieve a sufficient level of data accuracy. The experiments are conducted in March and April within the selected conditions as described in detail to avoid unexpected interference from other factors. The CO2 level has some relation to the changes in T and RH in this work; however, in general, the CO2 may also be affected by other factors, such as human occupation, ventilation through windows, air con,… Therefore, the results from this work are difficult to apply to other situations. Even though the authors have mentioned the limitation and future works of more tests to be done, but you should discuss more in details how to scale this works, for example through some other finding/principle of the relationship between these three parameters (will need more references or background to be explained in the Introduction section), or any models can be used to predict these threshold trigger values/ranges. With the current presentation in this manuscript, I may understand that the tests need to be done for each situation to be used with this proposed 2-to-1 sampling approach, which is not very feasible in terms of time and cost. There are so many factors that could change the threshold values of T and RH, how would the authors like to deal with this?

Response: We agree that the relationship between temperature (T), relative humidity (RH) and CO2 is context-dependent and depends on a number of external factors such as human presence, ventilation, room space, etc. Therefore, fixed thresholds are unlikely to be optimal for all settings.

BUT, to overcome this limitation, we are exploring the following strategies in future work:

Local calibration step : in future work we will propose an initial calibration step before deployment, during which the sensor system collects data to determine environment-specific correlations between T, RH and CO2. This calibration can be used to automatically adjust thresholds using statistical methods (e.g., moving averages, correlation coefficients) without the need for manual tuning. This approach will allow the method to adapt to different environments without the need for full re-testing for each new scenario.

We have also updated the Limitations section where we explicitly state that the thresholds used in the current study were empirically derived from specific conditions in March and April and that they may not be generalizable to other environments or seasons.

1.b.

Comment: The 2-to-1 method was implemented only on the PIC platform, but the Arduino and Loxone systems were used as reference points for comparison. This helped validate the accuracy of our method under the same environmental conditions.

Response: Yes. I updated the section “Materials” and have been added at the end of the section the text starting with “The proposed method was implemented”

1.c.

Comment: All three sensor platforms were deployed under the same environmental conditions and synchronized sampling intervals (every 15minutes) to ensure fair comparison. The observed weak correlation in Sensor 1 is likely due to hardware limitations (lower resolution and sensitivity) rather than methodological inconsistencies.

Nevertheless, the proposed 2-to-1 threshold-based sampling logic does not rely on strong inter-sensor correlation. Instead, it is based on localized threshold exceedance within the same dataset, making it robust to hardware variability. This ensures that the trigger remains functional even when the overall statistical correlation is low, as local signal variations still activate the measurement logic reliably.

Response:

I agree that the weak correlation observed in Sensor 1 is likely due to its hardware characteristics, including lower resolution and sensitivity compared to other platforms. It is important to note here that the proposed 2-to-1 threshold-based sampling logic does not rely on global correlation between parameters across devices. Instead, it relies on local threshold exceedances within a single sensor data set. This ensures that the triggering mechanism remains functional even in cases where the overall statistical correlation is low, since local parameter changes still reliably trigger the CO2 measurement.

And I have added this at the end of the Correlation section, starting with: 'Although the weak correlation observed in Sensor 1 may indicate...'

1.d.

Comment: The comparisons are fair because all sensors were deployed in similar types of indoor environments with controlled conditions, including roughly similar room sizes, usage patterns, and ventilation settings. Although Sensor 2 collected data in April, environmental conditions were kept constant to minimize seasonal variations.

Response:

Yes, sensor 2 collected data in April to maintain constant environmental conditions to minimize the impact of seasonal variations.

And I updated the text and added this in \subsection{Data Acquisition}

starting with “Although Sensor 2 recorded data in April, this time window was“

1.e.

Comment: The main purpose of using three different systems was not to implement a 2:1 approach on all platforms, but to demonstrate the practicality, efficiency, and compactness of the PIC16-based implementation compared to standard (Arduino) and commercial (Loxone) solutions. While only the PIC based sensor used a 2:1 threshold approach, the data from the other two systems served as full resolution baselines to evaluate signal fidelity, confirm data integrity, and illustrate tradeoffs in power use and device form factor.

Response: Yes the main purpose of using three different systems was not to apply the 2 to 1 threshold approach on all platforms, but to demonstrate the practicality, efficiency and compactness of the PIC16 based implementation compared to standard (Arduino) and commercial (Loxone) solutions.

I’ve update the text in \subsection{Trigger Justification} starting with “Although the proposed 2-to-1 triggering logic was only implemented on”

1.f.

Comment: What is the purpose to review the data reduction methods in section 2.1.

At the end, the author did not compare the results from proposed method

with any of these baseline methods.

Response: The purpose of including this list was to provide a broader context for the data compression methods commonly used in data processing (often in signal processing).

The proposed method focuses specifically on conditional activation based on interparameter thresholds, which is fundamentally different in logic from classical data compression methods. Therefore, no direct comparison was made since the goals and working assumptions of the proposed approach are different.

But, yes, we accepted that the text lacked a clear explanation of why these methods were given. And the explanation was added at the end of Section 2.1 after the table with given methods to highlight the contrast with our threshold-based sampling method, which is specifically developed for low power microcontroller.

1.g.

Comment: There is no information or discussion of the saving of energy usage or

compact installation on this proposed method.

In the Results section I described the energy and installation implications, highlighting the reduction in CO2 emissions and the use of a small form factor microcontroller platform.

It has been added in section Discussion, it starts as “The observed 41.9% reduction in the $CO_2$”

  1. a.

Comment: A serious presentation when presenting the sensors systems 1, 2, and 3 that making the works are extremely difficult to follow. In the introduction, last sentence, the sensor 1 is Loxone system, sensor 2 is PIC16 based, sensor 3 is Arduino based. Then in section 2, figure 3, sensor 1 is now PIC16 based, sensor 2 is Arduino, sensor 3 is Loxone. Subsequently, the order of data in Figures 7 and 9 are Arduino, Loxone, then PIC16 based. This presentation, and if unexpected typo caused confusing when I try to understand the text when the authors compare the data from each sensor throughout the whole manuscript. A good practice is making them all in consistent names and orders throughout the whole manuscript, whatever they are in texts, tables, or figures.

Response: For clarity and consistency, we have revised the manuscript and are now named as follows:

Sensor 1: Custom PIC16LF19156 based sensor

Sensor 2: Arduino Uno based sensor

Sensor 3: Commercial Loxone system

And this clarification has been added at the beginning of the Materials and Methods sections.

2.b. Comment: In section 2.3 in data acquisition, if the data of all three sensors or any two

sensors are collected at the same time period, the data can be plotted

together to provide a clear visualization for comparison purposes.

Response: The recommendation, we have added combined visualizations for the sensors that were measuring during the same time period (Sensor 1 and Sensor 2 in March).

2.c. Comment: In figure 5, sensor 3 data is on March in the x axis but title is on April. The

similar mistake on figure 6.

Response: Yes, it has been fixed.

Round 2

Reviewer 1 Report

Comments and Suggestions for Authors

The author has provided a good response to the previous question, and the paper has made significant improvements. However, there are still some issues that need further modification and explanation:
1. The font in Figure 8 is too small to see clearly.
2. What is the relationship between Figure 4, Figure 10, and Figure 11? How to reconstruct data.
3. The reference citation in the revised manuscript is incorrect, that is, the 34th reference should be:Engineering Structures, Vol. 327, No. 3, 119523.

Author Response

Comment 1: The font in Figure 8 is too small to see clearly.
Response 1: Yes, the font has been fixed.

Comment 2: What is the relationship between Figure 4, Figure 10, and Figure 11? How to reconstruct data.
Response 2: I have update the text a bit and it can be found in "This visualization allows us to identify"

Figure 4 shows the raw environmental data measured by Sensor 1 and Sensor 3 in March, which forms the basis for threshold selection. Figure 10 builds on this by presenting an F1-score heatmap showing which threshold combinations best balance the precision and recall of CO$_2$ peak detection, allowing the selection of optimal trigger conditions for data processing. Then these thresholds are applied to the data to activate the $CO_2$ measurement, with the resulting reconstructed signal obtained from the reduced set of measurement points shown in figure 11.

Regarding how the data is reconstructed:
The reconstruction is done by using a reduced set of measurement points that were identified when temperature and humidity thresholds were exceeded.

I added in the Result section starting with "Figures \ref{fig:sensor_measurement_loxone_arduino}, \ref{fig:sensor_measurement7}, \ref{fig:heat_map_result}, and \ref{fig:reconstructed} together present..."

Comment 3: The reference citation in the revised manuscript is incorrect, that is, the 34th reference should be:Engineering Structures, Vol. 327, No. 3, 119523.
Response 3: The correct citation has been updated and can be found here "These effects can contribute to long-term degradation and structural responses, as shown in recent studies \cite{Ding2025}"

Reviewer 2 Report

Comments and Suggestions for Authors

Thank you for your responses. The revised manuscript fully addressed all of my concerns.

Author Response

Comments 1: Thank you for your responses. The revised manuscript fully addressed all of my concerns.

Response 2: Thank you for your feedback and support.